# MERS-CoV spillover at the camel-human interface

Gytis Dudas[1]*, Luiz Max Carvalho[2], Andrew Rambaut[2,3], Trevor Bedford[1]

[1]Vaccine and Infectious Disease Division, Fred Hutchinson Cancer Research Center, Seattle, United States; [2]Institute of Evolutionary Biology, University of Edinburgh, Edinburgh, United Kingdom; [3]Fogarty International Center, National Institutes of Health, Bethesda, United States

**Abstract** Middle East respiratory syndrome coronavirus (MERS-CoV) is a zoonotic virus from camels causing significant mortality and morbidity in humans in the Arabian Peninsula. The epidemiology of the virus remains poorly understood, and while case-based and seroepidemiological studies have been employed extensively throughout the epidemic, viral sequence data have not been utilised to their full potential. Here, we use existing MERS-CoV sequence data to explore its phylodynamics in two of its known major hosts, humans and camels. We employ structured coalescent models to show that long-term MERS-CoV evolution occurs exclusively in camels, whereas humans act as a transient, and ultimately terminal host. By analysing the distribution of human outbreak cluster sizes and zoonotic introduction times, we show that human outbreaks in the Arabian peninsula are driven by seasonally varying zoonotic transfer of viruses from camels. Without heretofore unseen evolution of host tropism, MERS-CoV is unlikely to become endemic in humans.

DOI: https://doi.org/10.7554/eLife.31257.001

## Introduction

Middle East respiratory syndrome coronavirus (MERS-CoV), endemic in camels in the Arabian Peninsula, is the causative agent of zoonotic infections and limited outbreaks in humans. The virus, first discovered in 2012 (*Zaki et al., 2012*; *van Boheemen et al., 2012*), has caused more than 2000 infections and over 700 deaths, according to the World Health Organization (WHO) (*World Health Organization, 2017*). Its epidemiology remains obscure, largely because infections are observed among the most severely affected individuals, such as older males with comorbidities (*Assiri et al., 2013a*; *WHO MERS-Cov Research Group, 2013*). While contact with camels is often reported, other patients do not recall contact with any livestock, suggesting an unobserved community contribution to the outbreak (*WHO MERS-Cov Research Group, 2013*). Previous studies on MERS-CoV epidemiology have used serology to identify factors associated with MERS-CoV exposure in potential risk groups (*Reusken et al., 2015*; *Reusken et al., 2013*). Such data have shown high seroprevalence in camels (*Müller et al., 2014*; *Corman et al., 2014*; *Chu et al., 2014*; *Reusken et al., 2013*; *Reusken et al., 2014*) and evidence of contact with MERS-CoV in workers with occupational exposure to camels (*Reusken et al., 2015*; *Müller et al., 2015*). Separately, epidemiological modelling approaches have been used to look at incidence reports through time, space and across hosts (*Cauchemez et al., 2016*).

Although such epidemiological approaches yield important clues about exposure patterns and potential for larger outbreaks, much inevitably remains opaque to such approaches due to difficulties in linking cases into transmission clusters in the absence of detailed information. Where sequence data are relatively cheap to produce, genomic epidemiological approaches can fill this critical gap in outbreak scenarios (*Liu et al., 2013*; *Gire et al., 2014*; *Grubaugh et al., 2017*). These

**\*For correspondence:**
gdudas@fredhutch.org

**Competing interests:** The authors declare that no competing interests exist.

**eLife digest** Coronaviruses are one of many groups of viruses that cause the common cold, though some members of the group can cause more serious illnesses. The SARS coronavirus, for example, caused a widespread epidemic of pneumonia in 2003 that killed 774 people. In 2012, a new coronavirus was detected in patients from the Arabian Peninsula with severe respiratory symptoms known as Middle East respiratory syndrome (or MERS for short). To date the MERS coronavirus has also killed over 700 people (albeit over a number of years rather than months). Yet unlike the SARS coronavirus that spreads efficiently between humans, cases of MERS were rarely linked to each other or to contact with animals, with the exception of hospital outbreaks.

Though camels were later identified as the original source of MERS coronavirus infections in humans, the role of these animals in the epidemic was not well understood. Throughout the epidemic nearly 300 genomes of the MERS coronavirus had been sequenced, from both camels and humans. Previous attempts to understand the MERS epidemic had either relied on these data or reports of case numbers but led to conflicting results, at odds with other sources of evidence.

Dudas et al. wanted to work out how many times the MERS coronavirus had been introduced into humans from camels. If it happened once, this would indicate that the virus is good at spreading between humans and that treating human cases should be a priority. However, if every human case occurred as a new introduction of the MERS coronavirus from camels, this would mean that the human epidemic would not stop until the virus is controlled at the source, that is, in camels. Many scientists had argued that the second of these scenarios was most likely, but this had not been strongly demonstrated with data.

By looking at the already sequenced genomes, Dudas et al. worked out how the MERS coronaviruses were related to each other, and reconstructed their family tree. Information about the host from which each sequence was collected was then mapped on the tree. Unlike previous attempts to complete this kind of analysis, Dudas et al. took an approach that could deal with the viruses in camels being more diverse than those in humans.

Consistent with the scenario where human cases occurred as new introductions from camels, the analysis showed that the MERS coronavirus populations is maintained exclusively in camels and the viruses seen in humans are evolutionary dead-ends. This suggests that MERS coronavirus spreads poorly between humans, and has instead jumped from camels to humans hundreds of times since 2012.

As well as providing data to confirm a previously suspected hypothesis, these findings provide more support to the current plans to mitigate infections with MERS coronavirus in the Arabian Peninsula by focusing control efforts on camels.

DOI: https://doi.org/10.7554/eLife.31257.002

data often yield a highly detailed picture of an epidemic when complete genome sequencing is performed consistently and appropriate metadata collected (*Dudas et al., 2017*). Sequence data can help discriminate between multiple and single source scenarios (*Gire et al., 2014*; *Quick et al., 2015*), which are fundamental to quantifying risk (*Grubaugh et al., 2017*). Sequencing MERS-CoV has been performed as part of initial attempts to link human infections with the camel reservoir (*Memish et al., 2014*), nosocomial outbreak investigations (*Assiri et al., 2013b*) and routine surveillance (*Wernery et al., 2015*). A large portion of MERS-CoV sequences come from outbreaks within hospitals, where sequence data have been used to determine whether infections were isolated introductions or were part of a larger hospital-associated outbreak (*Fagbo et al., 2015*). Similar studies on MERS-CoV have taken place at broader geographic scales, such as cities (*Cotten et al., 2013*).

It is widely accepted that recorded human MERS-CoV infections are a result of at least several introductions of the virus into humans (*Cotten et al., 2013*) and that contact with camels is a major risk factor for developing MERS, per WHO guidelines (*World Health Organization, 2016*). Previous studies attempting to quantify the actual number of spillover infections have either relied on case-based epidemiological approaches (*Cauchemez et al., 2016*) or employed methods agnostic to signals of population structure within sequence data (*Zhang et al., 2016*). Here, we use a dataset of

274 MERS-CoV genomes to investigate transmission patterns of the virus between humans and camels.

Here, we use an explicit model of metapopulation structure and migration between discrete sub-populations, referred to here as demes (*Vaughan et al., 2014*), derived from the structured coalescent (*Notohara, 1990*). Unlike approaches that model host species as a discrete phylogenetic trait of the virus using continuous-time Markov processes (or simpler, parsimony based, approaches) (*Faria et al., 2013*; *Lycett et al., 2016*), population structure models explicitly incorporate contrasts in deme effective population sizes and migration between demes. By estimating independent coalescence rates for MERS-CoV in humans and camels, as well as migration patterns between the two demes, we show that long-term viral evolution of MERS-CoV occurs exclusively in camels. Our results suggest that spillover events into humans are seasonal and might be associated with the calving season in camels. However, we find that MERS-CoV, once introduced into humans, follows transient transmission chains that soon abate. Using Monte Carlo simulations we show that $R_0$ for MERS-CoV circulating in humans is much lower than the epidemic threshold of 1.0 and that correspondingly the virus has been introduced into humans hundreds of times.

## Results

### MERS-CoV is predominantly a camel virus

The structured coalescent approach we employ (see Materials and methods) identifies camels as a reservoir host where most of MERS-CoV evolution takes place (*Figure 1*), while human MERS outbreaks are transient and terminal with respect to long-term evolution of the virus (*Figure 1—figure supplement 1*). Across 174 MERS-CoV genomes collected from humans, we estimate a median of 56 separate camel-to-human cross-species transmissions (95% highest posterior density (HPD): 48–63). While we estimate a median of 3 (95% HPD: 0–12) human-to-camel migrations, the 95% HPD interval includes zero and we find that no such migrations are found in the maximum clade credibility tree (*Figure 1*). Correspondingly, we observe substantially higher camel-to-human migration rate estimates than human-to-camel migration rate estimates (*Figure 1—figure supplement 2*). This inference derives from the tree structure wherein human viruses appear as clusters of highly related sequences nested within the diversity seen in camel viruses, which themselves show significantly higher diversity and less clustering. This manifests as different rates of coalescence with camel viruses showing a scaled effective population size $N_e\tau$ of 3.49 years (95% HPD: 2.71–4.38) and human viruses showing a scaled effective population of 0.24 years (95% HPD: 0.14–0.34).

We believe that the small number of inferred human-to-camel migrations are induced by the migration rate prior, while some are derived from phylogenetic proximity of human sequences to the apparent 'backbone' of the phylogenetic tree. This is most apparent in lineages in early-mid 2013 that lead up to sequences comprising the MERS-CoV clade dominant in 2015, where owing to poor sampling of MERS-CoV genetic diversity from camels the model cannot completely dismiss humans as a potential alternative host. The first sequences of MERS-CoV from camels do not appear in our data until November 2013. Our finding of negligible human-to-camel transmission is robust to choice of prior (*Figure 1—figure supplement 2*).

The repeated and asymmetric introductions of short-lived clusters of MERS-CoV sequences from camels into humans leads us to conclude that MERS-CoV epidemiology in humans is dominated by zoonotic transmission (*Figure 1* and *Figure 1—figure supplement 1*). We observe dense terminal clusters of MERS-CoV circulating in humans that are of no subsequent relevance to the evolution of the virus. These clusters of presumed human-to-human transmission are then embedded within extensive diversity of MERS-CoV lineages inferred to be circulating in camels, a classic pattern of source-sink dynamics. Our findings suggest that instances of human infection with MERS-CoV are more common than currently thought, with exceedingly short transmission chains mostly limited to primary cases that might be mild and ultimately not detected by surveillance or sequencing. Structured coalescent analyses recover the camel-centered picture of MERS-CoV evolution despite sequence data heavily skewed towards non-uniformly sampled human cases and are robust to choice of prior. Comparing these results with a currently standard discrete trait analysis (*Lemey et al., 2009*) approach for ancestral state reconstruction shows dramatic differences in host reconstruction at internal nodes (*Figure 1—figure supplement 3*). Discrete trait analysis reconstruction identifies

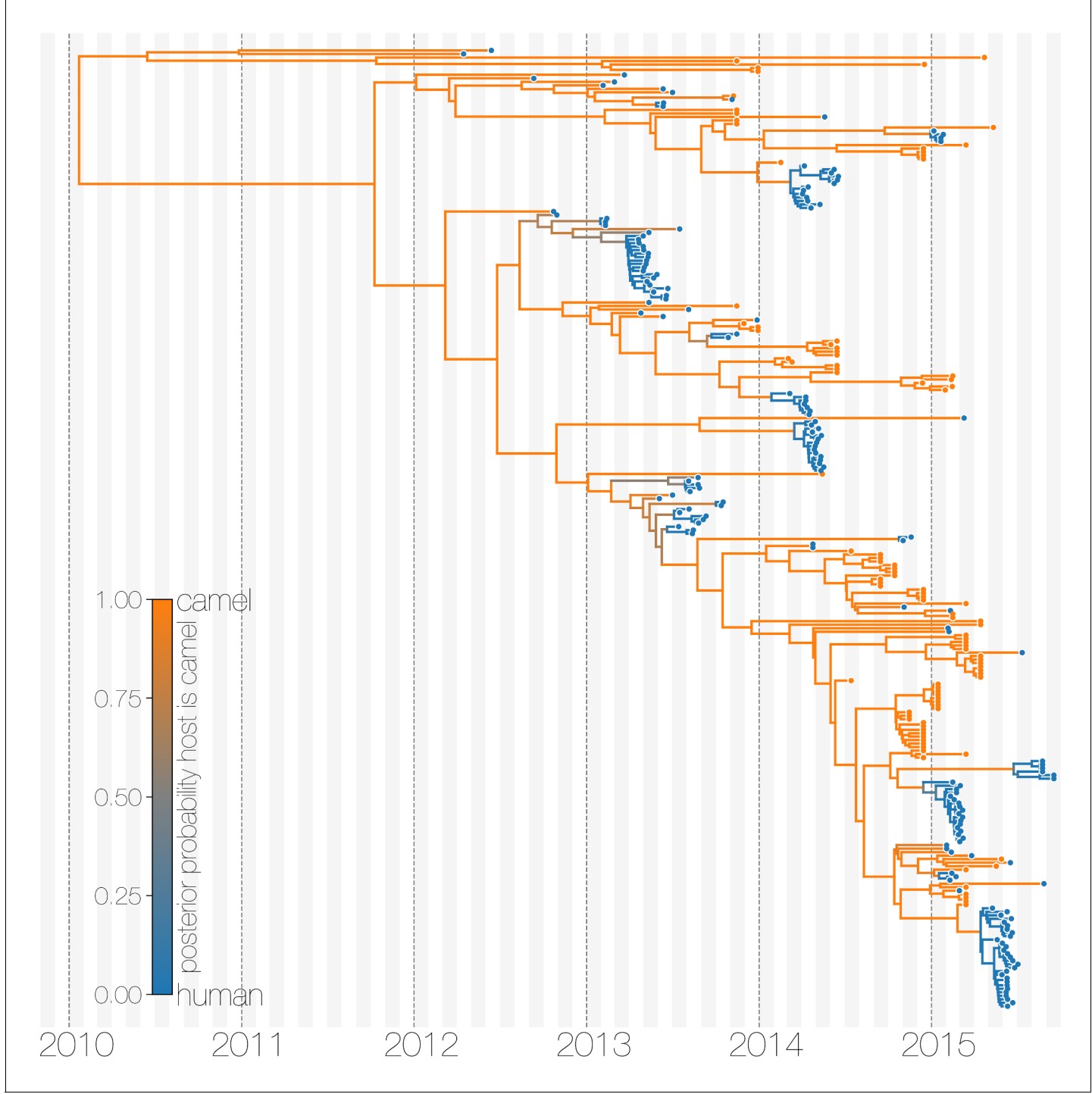

**Figure 1.** Typed maximum clade credibility tree of MERS-CoV genomes from 174 human viruses and 100 camel viruses. Maximum clade credibility (MCC) tree showing inferred ancestral hosts for MERS-CoV recovered with the structured coalescent. The vast majority of MERS-CoV evolution is inferred to occur in camels (orange) with human outbreaks (blue) representing evolutionary dead-ends for the virus. Confidence in host assignment is depicted as a colour gradient, with increased uncertainty in host assignment (posterior probabilities close to 0.5) shown as grey. While large clusters of human cases are apparent in the tree, significant contributions to human outbreaks are made by singleton sequences, likely representing recent cross-species transmissions that were caught early.

DOI: https://doi.org/10.7554/eLife.31257.003

The following source data and figure supplements are available for figure 1:

**Source data 1.** XML to run structured coalescent analysis and output files.

*Figure 1 continued on next page*

*Figure 1 continued*

DOI: https://doi.org/10.7554/eLife.31257.009

**Source data 2.** XML to run structured coalescent analysis with a relaxed prior and output file.

DOI: https://doi.org/10.7554/eLife.31257.010

**Source data 3.** XML to run discrete trait analysis (DTA) and output files.

DOI: https://doi.org/10.7554/eLife.31257.011

**Source data 4.** XML to run structured coalescent analysis with equal deme sizes between humans and camels and output files.

DOI: https://doi.org/10.7554/eLife.31257.012

**Source data 5.** Maximum likelihood phylogeny.

DOI: https://doi.org/10.7554/eLife.31257.013

**Figure supplement 1.** Evolutionary history of MERS-CoV partitioned between camels and humans.

DOI: https://doi.org/10.7554/eLife.31257.004

**Figure supplement 2.** Posterior backwards migration rate estimates for two choices of prior.

DOI: https://doi.org/10.7554/eLife.31257.005

**Figure supplement 3.** Maximum clade credibility (MCC) tree with ancestral state reconstruction according to a discrete trait model.

DOI: https://doi.org/10.7554/eLife.31257.006

**Figure supplement 4.** Maximum clade credibility (MCC) tree of structured coalescent model with enforced equal coalescence rates.

DOI: https://doi.org/10.7554/eLife.31257.007

**Figure supplement 5.** Maximum likelihood (ML) tree of MERS-CoV genomes coloured by origin of sequence.

DOI: https://doi.org/10.7554/eLife.31257.008

both camels and humans as important hosts for MERS-CoV persistence, but with humans as the ultimate source of camel infections. A similar approach has been attempted previously (*Zhang et al., 2016*), but this interpretation of MERS-CoV evolution disagrees with lack of continuing human transmission chains outside of Arabian peninsula, low seroprevalence in humans and very high seroprevalence in camels across Saudi Arabia. We suspect that this particular discrete trait analysis reconstruction is false due to biased data, that is, having nearly twice as many MERS-CoV sequences from humans ($n = 174$) than from camels ($n = 100$) and the inability of the model to account for and quantify vastly different rates of coalescence in the phylogenetic vicinity of both types of sequences. We can replicate these results by either applying the same model to current data (*Figure 1—figure supplement 3*) or by enforcing equal coalescence rates within each deme in the structured coalescent model (*Figure 1—figure supplement 4*).

## MERS-CoV shows seasonal introductions

We use the posterior distribution of MERS-CoV introduction events from camels to humans (*Figure 1*) to model seasonal variation in zoonotic transfer of viruses. We identify four months (April, May, June, July) when the odds of MERS-CoV introductions are increased (*Figure 2*) and four when the odds are decreased (August, September, November, December). Camel calving is reported to occur from October to February (*Almutairi et al., 2010*), with rapidly declining maternal antibody levels in calves within the first weeks after birth (*Wernery, 2001*). It is possible that MERS-CoV sweeps through each new camel generation once critical mass of susceptibles is reached (*Martinez-Bakker et al., 2014*), leading to a sharp rise in prevalence of the virus in camels and resulting in increased force of infection into the human population. Strong influx of susceptibles and subsequent sweeping outbreaks in camels may explain evidence of widespread exposure to MERS-CoV in camels from seroepidemiology (*Müller et al., 2014*; *Corman et al., 2014*; *Chu et al., 2014*; *Reusken et al., 2013*; *Reusken et al., 2014*).

Although we find evidence of seasonality in zoonotic spillover timing, no such relationship exists for sizes of human sequence clusters (*Figure 2B*). This is entirely expected, since little seasonality in human behaviour that could facilitate MERS-CoV transmission is expected following an introduction. Similarly, we do not observe any trend in human sequence cluster sizes over time (*Figure 2B*, Spearman $\rho = 0.06$, $p = 0.68$), suggesting that MERS-CoV outbreaks in humans are neither growing nor shrinking in size. This is not surprising either, since MERS-CoV is a camel virus that has to date, experienced little-to-no selective pressure to improve transmissibility between humans.

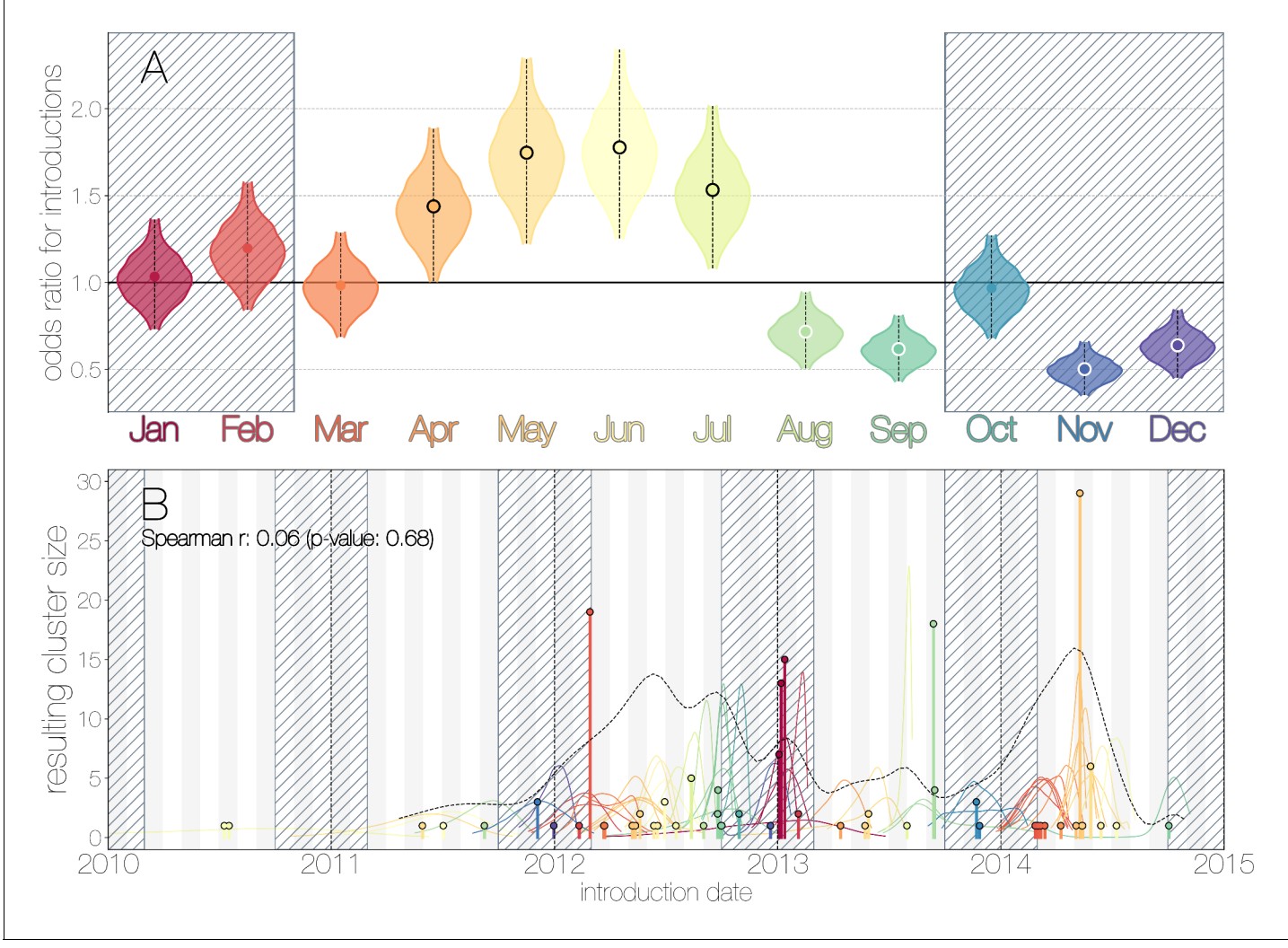

**Figure 2.** Seasonality of MERS-CoV introduction events. (A) Posterior density estimates partitioned by month showing the 95% highest posterior density interval for relative odds ratios of MERS-CoV introductions into humans. Posterior means are indicated with circles. Evidence for increased or decreased risk (95% HPD excludes 1.0) for introductions are indicated by black or white circles, respectively. Hatched area spanning October to February indicates the camel calving season. (B) Sequence cluster sizes and inferred dates of introduction events. Each introduction event is shown as a vertical line positioned based on the median introduction time, as recovered by structured coalescent analyses and coloured by time of year with height indicating number of descendant sequences recovered from human cases. 95% highest posterior density intervals for introductions of MERS-CoV into humans are indicated with coloured lines, coloured by median estimated introduction time. The black dotted line indicates the joint probability density for introductions. We find little correlation between date and size of introduction (Spearman $\rho = 0.06$, $p = 0.68$).

DOI: https://doi.org/10.7554/eLife.31257.014

The following source data is available for figure 2:

**Source data 1.** MCMC samples from seasonality inference analysis.

DOI: https://doi.org/10.7554/eLife.31257.015

## MERS-CoV is poorly suited for human transmission

Structured coalescent approaches clearly show humans to be a terminal host for MERS-CoV, implying poor transmissibility. However, we wanted to translate this observation into an estimate of the basic reproductive number $R_0$ to provide an intuitive epidemic behaviour metric that does not require expertise in reading phylogenies. The parameter $R_0$ determines expected number of secondary cases in a single infections as well as the distribution of total cases that result from a single introduction event into the human population (*Equation 1*, Materials and methods). We estimate $R_0$ along with other relevant parameters via Monte Carlo simulation in two steps. First, we simulate

case counts across multiple outbreaks totaling 2000 cases using *Equation 1* and then we subsample from each case cluster to simulate sequencing of a fraction of cases. Sequencing simulations are performed via a multivariate hypergeometric distribution, where the probability of sequencing from a particular cluster depends on available sequencing capacity (number of trials), numbers of cases in the cluster (number of successes) and number of cases outside the cluster (number of failures). In addition, each hypergeometric distribution used to simulate sequencing is concentrated via a bias parameter, that enriches for large sequence clusters at the expense of smaller ones (for its effects see *Figure 3—figure supplement 1*). This is a particularly pressing issue, since *a priori* we expect large hospital outbreaks of MERS to be overrepresented in sequence data, whereas sequences from primary cases will be sampled exceedingly rarely. We record the number, mean, standard deviation and skewness (third standardised moment) of sequence cluster sizes in each simulation (left-hand panel in *Figure 3*) and extract the subset of Monte Carlo simulations in which these summary statistics fall within the 95% highest posterior density observed in the empirical MERS-CoV data from structured coalescent analyses. We record $R_0$ values, as well as the number of case clusters (equivalent to number of zoonotic introductions), for these empirically matched simulations. A schematic of this Monte Carlo procedure is shown in *Figure 3—figure supplement 2*. Since the total number of

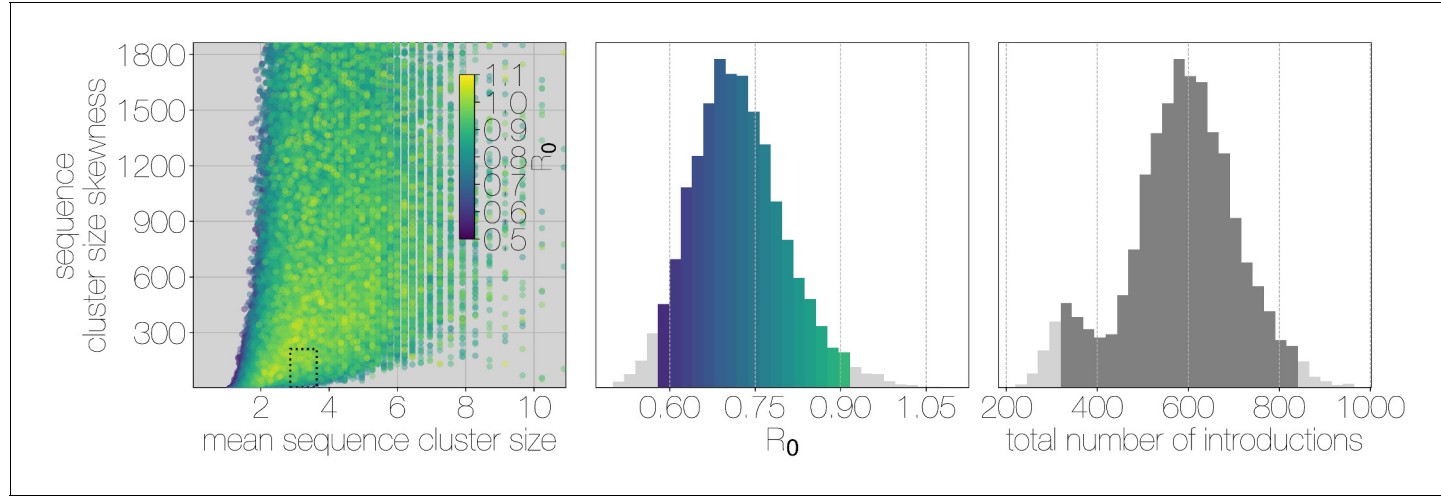

**Figure 3.** Monte Carlo simulations of human transmission clusters. Leftmost scatter plot shows the distribution of individual Monte Carlo simulation sequence cluster size statistics (mean and skewness) coloured by the $R_0$ value used for the simulation. The dotted rectangle identifies the 95% highest posterior density bounds for sequence cluster size mean and skewness observed for empirical MERS-CoV data. The distribution of $R_0$ values that fall within 95% HPDs for sequence cluster size mean, standard deviation, skewness and number of introductions, is shown in the middle, on the same *y*-axis. Bins falling inside the 95% percentiles are coloured by $R_0$, as in the leftmost scatter plot. The distribution of total number of introductions associated with simulations matching MERS-CoV sequence clusters is shown on the right. Darker shade of grey indicates bins falling within the 95% percentiles. Monte Carlo simulations indicate $R_0$ for MERS-CoV in humans is likely to be below 1.0, with numbers of zoonotic transmissions numbering in the hundreds.

DOI: https://doi.org/10.7554/eLife.31257.016

The following figure supplements are available for figure 3:

**Figure supplement 1.** Monte Carlo simulations of human transmission clusters.
DOI: https://doi.org/10.7554/eLife.31257.017

**Figure supplement 2.** Monte Carlo simulation schematic.
DOI: https://doi.org/10.7554/eLife.31257.018

**Figure supplement 3.** Results of Monte Carlo simulations with vast underestimation of cases.
DOI: https://doi.org/10.7554/eLife.31257.019

**Figure supplement 4.** Boxplots of matching simulated case and sequence cluster distributions.
DOI: https://doi.org/10.7554/eLife.31257.020

**Figure supplement 5.** Quantile-quantile (Q-Q) plot of empirical and simulated sequence cluster sizes.
DOI: https://doi.org/10.7554/eLife.31257.021

**Figure supplement 6.** Numbers of epidemiological simulations conforming to empirical observations.
DOI: https://doi.org/10.7554/eLife.31257.022

cases is fixed at 2000, higher $R_0$ results in fewer larger transmission clusters, while lower $R_0$ results in many smaller transmission clusters.

We find that observed phylogenetic patterns of sequence clustering strongly support $R_0$ below 1.0 (middle panel in *Figure 3*). Mean $R_0$ value observed in matching simulations is 0.72 (95% percentiles 0.57–0.90), suggesting the inability of the virus to sustain transmission in humans. Lower values for $R_0$ in turn suggest relatively large numbers of zoonotic transfers of viruses into humans (right-hand panel in *Figure 3*). Median number of cross-species introductions observed in matching simulations is 592 (95% percentiles 311–811). Our results suggest a large number of unobserved MERS primary cases. Given this, we also performed simulations where the total number of cases is doubled to 4000 to explore the impact of dramatic underestimation of MERS cases. In these analyses, $R_0$ values tend to decrease even further as numbers of introductions go up, although very few simulations match currently observed MERS-CoV sequence data (*Figure 3—figure supplement 3*).

Overall, our analyses indicate that MERS-CoV is poorly suited for human-to-human transmission, with an estimated $R_0 < 0.90$ and sequence sampling likely to be biased towards large hospital outbreaks (*Figure 3—figure supplement 1*). All matching simulations exhibit highly skewed distributions of case cluster sizes with long tails (*Figure 3—figure supplement 4*), which is qualitatively similar to the results of (*Cauchemez et al., 2016*). We find that simulated sequence cluster sizes resemble observed sequence cluster sizes in the posterior distribution, with a slight reduction in mid-sized clusters in simulated data (*Figure 3—figure supplement 5*). Given these findings, and the fact that large outbreaks of MERS occurred in hospitals, the combination of frequent spillover of MERS-CoV into humans and occasional outbreak amplification via poor hygiene practices (*Assiri et al., 2013b*; *Chen et al., 2017*) appear sufficient to explain observed epidemiological patterns of MERS-CoV.

## Recombination shapes MERS-CoV diversity

Recombination has been shown to occur in all genera of coronaviruses, including MERS-CoV (*Lai et al., 1985*; *Makino et al., 1986*; *Keck et al., 1988*; *Kottier et al., 1995*; *Herrewegh et al., 1998*). In order to quantify the degree to recombination has shaped MERS-CoV genetic diversity, we used two recombination detection approaches across partitions of taxa corresponding to inferred MERS-CoV clades. Both methods rely on sampling parental and recombinant alleles within the alignment, although each quantifies different signals of recombination. One hallmark of recombination is the ability to carry alleles derived via mutation from one lineage to another, which appear as repeated mutations taking place in the recipient lineage, somewhere else in the tree. The PHI (pairwise homoplasy index) test quantifies the appearance of these excessive repeat mutations (homoplasies) within an alignment (*Bruen et al., 2006*). Another hallmark of recombination is clustering of alleles along the genome, due to how template switching, the primary mechanism of recombination in RNA viruses, occurs. 3Seq relies on the clustering of nucleotide similarities along the genome between sequence triplets – two potential parent-donors and one candidate offspring-recipient sequences (*Boni et al., 2007*).

Both tests can give spurious results in cases of extreme rate heterogeneity and sampling over time (*Dudas and Rambaut, 2016*), but both tests have not been reported to fail simultaneously. PHI and 3Seq methods consistently identify most of the apparent 'backbone' of the MERS-CoV phylogeny as encompassing sequences with evidence of recombination (*Figure 4—figure supplement 1*). Neither method can identify where in the tree recombination occurred, but each full asterisk in *Figure 4—figure supplement 1* should be interpreted as the minimum partition of data that still captures both donor and recipient alleles involved in a recombination event. This suggests a non-negligible contribution of recombination in shaping existing MERS-CoV diversity. As done previously (*Dudas and Rambaut, 2016*), we show large numbers of homoplasies in MERS-CoV data (*Figure 4—figure supplement 2*) with some evidence of genomic clustering of such alleles. These results are consistent with high incidence of MERS-CoV in camels (*Müller et al., 2014*; *Corman et al., 2014*; *Chu et al., 2014*; *Reusken et al., 2014*; *Ali et al., 2017*), allowing for co-infection with distinct genotypes and thus recombination to occur (*Sabir et al., 2016*).

Owing to these findings, we performed a sensitivity analysis in which we partitioned the MERS-CoV genome into two fragments and identified human outbreak clusters within each fragment. We find strong similarity in the grouping of human cases into outbreak clusters between fragments (*Figure 4A*). Between the two trees in *Figure 4B* four (out of 54) 'human' clades are expanded

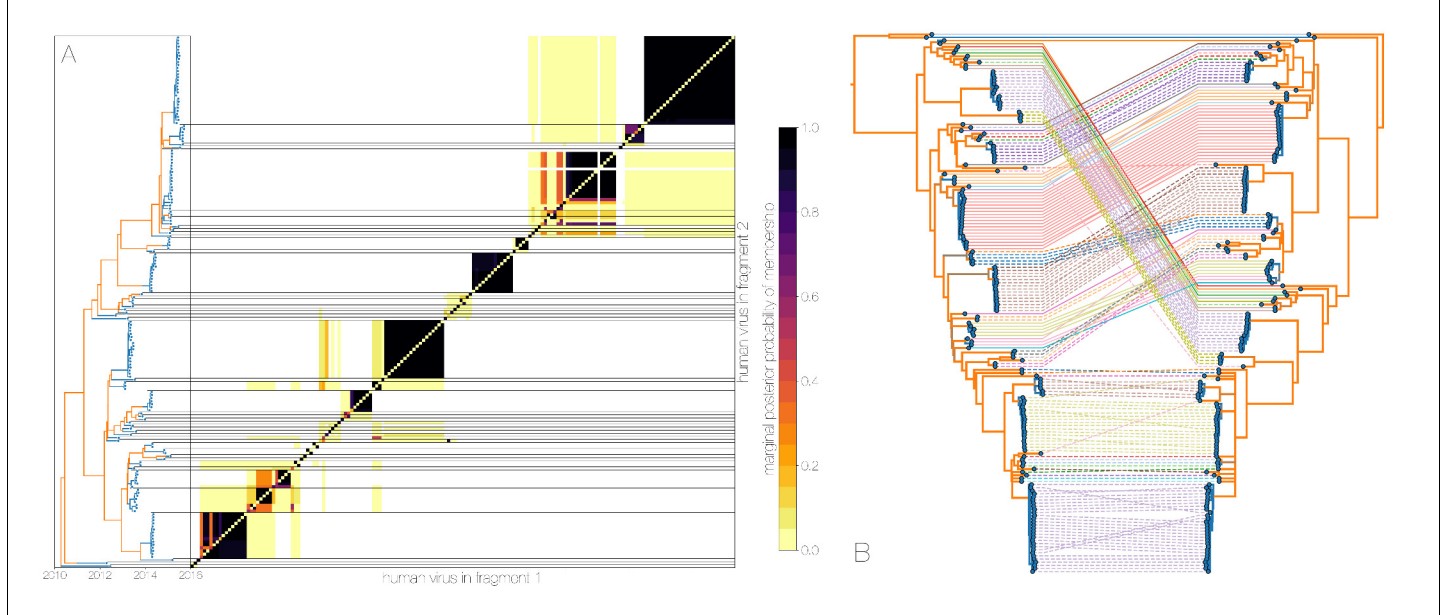

**Figure 4.** Recombinant features of MERS-CoV phylogenies. (**A**) Marginal posterior probabilities of taxa collected from humans belonging to the same clade in phylogenies derived from different parts of the genome. Taxa are ordered according to phylogeny of fragment 2 (genome positions 21001 to 29364) reduced to just the human tips and is displayed on the left. Human clusters are largely well-supported as monophyletic and consistent across trees of both genomic fragments. (**B**) Tanglegram connecting the same taxa between a phylogeny derived from fragment 1 (left, genome positions 1 to 21000) and fragment 2 (right, genome positions 21001 to 29364), reduced to just the human tips and branches with posterior probability <0.1 collapsed. Human clusters exhibit limited diversity and corresponding low levels of incongruence within an introduction cluster.

DOI: https://doi.org/10.7554/eLife.31257.023

The following source data and figure supplements are available for figure 4:

**Source data 1.** XML to run structured coalescent analysis on bisected alignment with output files.
DOI: https://doi.org/10.7554/eLife.31257.027
**Source data 2.** Output from PHI and 3Seq recombination analyses.
DOI: https://doi.org/10.7554/eLife.31257.028
**Source data 3.** Output from ClonalFrameML analysis.
DOI: https://doi.org/10.7554/eLife.31257.029
**Figure supplement 1.** Tests of recombination across MERS-CoV clades.
DOI: https://doi.org/10.7554/eLife.31257.024
**Figure supplement 2.** MERS-CoV genomes exhibit high numbers of non-clonal loci.
DOI: https://doi.org/10.7554/eLife.31257.025
**Figure supplement 3.** Human clade sharing between genomic fragments 1 and 2.
DOI: https://doi.org/10.7554/eLife.31257.026

where either singleton introductions or two-taxon clades in fragment 2 join other clades in fragment 1. For the reverse comparison, there are five 'human' clades (out of 53) in fragment 2 that are expanded. All such clades have low divergence (*Figure 4B*) and thus incongruences in human clades are more likely to be caused by differences in deme assignment rather than actual recombination. And while we observe evidence of distinct phylogenetic trees from different parts of the MERS-CoV genome (*Figure 4B*), human clades are minimally affected and large portions of the posterior probability density in both parts of the genome are concentrated in shared clades (*Figure 4—figure supplement 3*). Additionally, we observe the same source-sink dynamics between camel and human populations in trees constructed from separate genomic fragments as were observed in the original full genome tree (*Figures 1* and *4B*).

Observed departures from strictly clonal evolution suggest that while recombination is an issue for inferring MERS-CoV phylogenies, its effect on the human side of MERS outbreaks is minimal, as expected if humans represent a transient host with little opportunity for co-infection. MERS-CoV evolution on the reservoir side is complicated by recombination, although is nonetheless still largely

amenable to phylogenetic methods. Amongst other parameters of interest, recombination is expected to interfere with molecular clocks, where transferred genomic regions can give the impression of branches undergoing rapid evolution, or branches where recombination results in reversions appearing to evolve slow. In addition to its potential to influence tree topology, recombination in molecular sequence data is an erratic force with unpredictable effects. We suspect that the effects of recombination in MERS-CoV data are reigned in by a relatively small effective population size of the virus in Saudi Arabia (see next section) where haplotypes are fixed or nearly fixed, thus preventing an accumulation of genetic diversity that would then be reshuffled via recombination. Nevertheless the evolutionary rate of the virus appears to fall firmly within the expected range for RNA viruses: regression of nucleotide differences to Jordan-N3/2012 genome against sequence collection dates yields a rate of $4.59 \times 10^{-4}$ subs/site/year, Bayesian structured coalescent estimate from primary analysis $9.57 \times 10^{-4}$ (95% HPDs: $8.28 - 10.9 \times 10^{-4}$) subs/site/year.

## MERS-CoV shows population turnover in camels

Here, we attempt to investigate MERS-CoV demographic patterns in the camel reservoir. We supplement camel sequence data with a single earliest sequence from each human cluster, treating viral diversity present in humans as a sentinel sample of MERS-CoV diversity circulating in camels. This removes conflicting demographic signals sampled during human outbreaks, where densely sampled closely related sequences from humans could be misconstrued as evidence of demographic crash in the viral population.

Despite lack of convergence, neither of the two demographic reconstructions show evidence of fluctuations in the scaled effective population size ($N_e\tau$) of MERS-CoV over time (*Figure 5*). We recover a similar demographic trajectory when estimating $N_e\tau$ over time with a skygrid tree prior across the genome split into ten fragments with independent phylogenetic trees to account for confounding effects of recombination (*Figure 5—figure supplement 1*). However, we do note that coalescence rate estimates are high relative to the sampling time period, with a mean estimate of $N_e\tau$ at 3.49 years (95% HPD: 2.71–4.38), and consequently MERS-CoV phylogeny resembles a ladder, as often seen in human influenza A virus phylogenies (*Bedford et al., 2011*).

This empirically estimated effected population can be compared to the expected effective population size in a simple epidemiological model. At endemic equilibrium, we expect scaled effective population size $N_e\tau$ to follow $I/2\beta$, where $\beta$ is the equilibrium rate of transmission and $I$ is the equilibrium number of infecteds (*Frost and Volz, 2010*). We assume that $\beta$ is constant and is equal to the rate of recovery. Given a 20 day duration of infection in camels (*Adney et al., 2014*), we arrive at $\beta = 365/20 = 18.25$ infections per year. Given extremely high seroprevalence estimates within camels in Saudi Arabia (*Müller et al., 2014*; *Corman et al., 2014*; *Chu et al., 2014*; *Reusken et al., 2013*; *Reusken et al., 2014*), we expect camels to usually be infected within their first year of life. Therefore, we can estimate the rough number of camel infections per year as the number of calves produced each year. We find there are 830,000 camels in Saudi Arabia (*Abdallah and Faye, 2013*) and that female camels in Saudi Arabia have an average fecundity of 45% (*Abdallah and Faye, 2013*). Thus, we expect $830\,000 \times 0.50 \times 0.45 = 186\,750$ new calves produced yearly and correspondingly 186,750 new infections every year, which spread over 20 day intervals gives an average prevalence of $I = 10\,233$ infections. This results in an expected scaled effective population size $N_e\tau = 280.4$ years.

Comparing expected $N_e\tau$ to empirical $N_e\tau$ we arrive at a ratio of 80.3 (64.0–103.5). This is less than the equivalent ratio for human measles virus (*Bedford et al., 2011*) and is in line with the expectation from neutral evolutionary dynamics plus some degree of transmission heterogeneity (*Volz et al., 2013*) and seasonal troughs in prevalence. Thus, we believe that the ladder-like appearance of the MERS-CoV tree can likely be explained by entirely demographic factors.

## Discussion

### MERS-CoV epidemiology

In this study we aimed to understand the drivers of MERS coronavirus transmission in humans and what role the camel reservoir plays in perpetuating the epidemic in the Arabian peninsula by using sequence data collected from both hosts (174 from humans and 100 from camels). We showed that

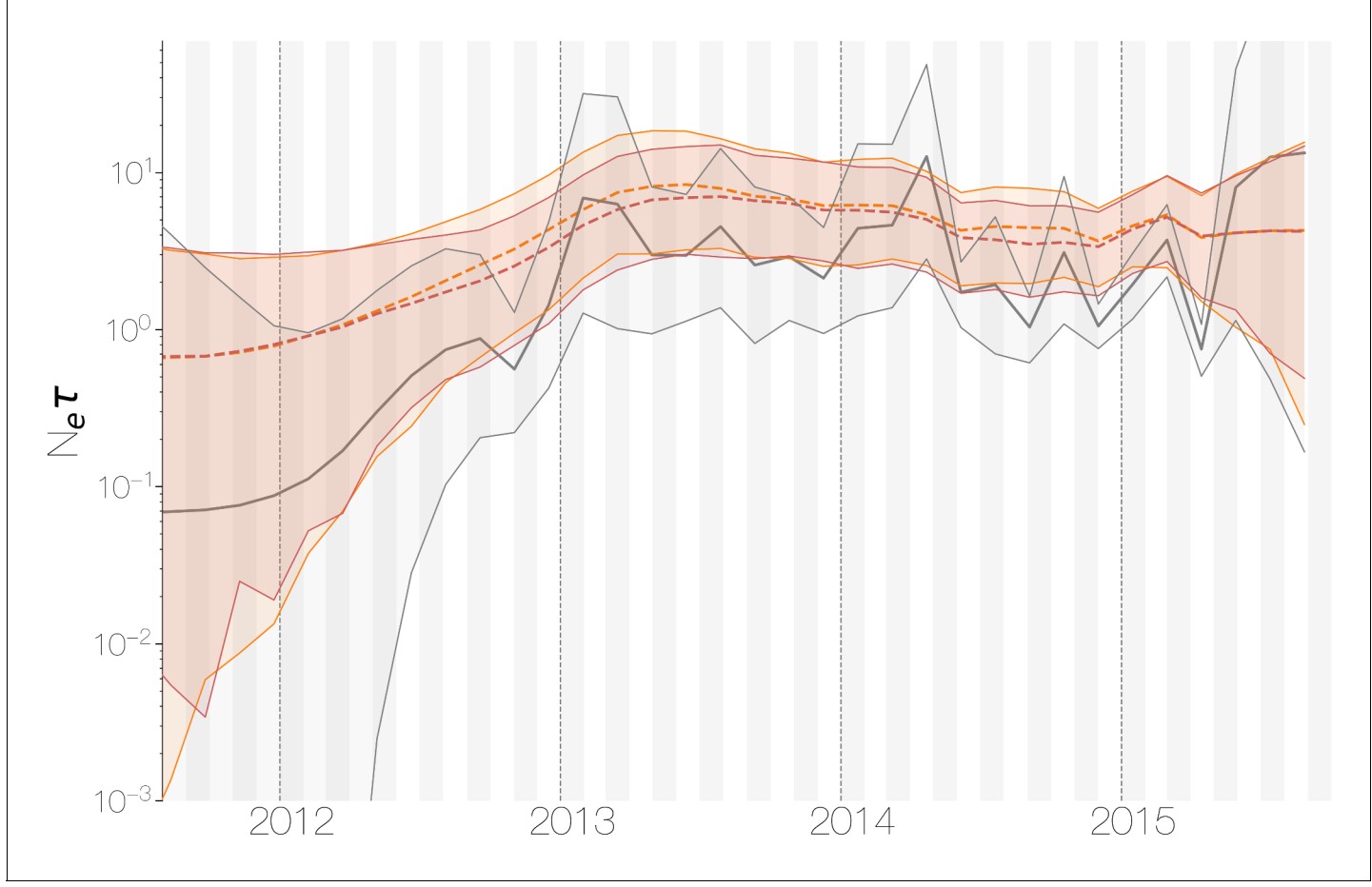

**Figure 5.** Demographic history of MERS-CoV in Arabian peninsula camels. Demographic history of MERS-CoV in camels, as inferred via a skygrid coalescent tree prior (*Gill et al., 2013*). Three skygrid reconstructions are shown, red and orange for each of the stationary distributions reached by MCMC with the whole genome and a black one where the genome was split into ten partitions. Shaded interval indicates the 95% highest posterior density interval for the product of generation time and effective population size, $N_e\tau$. Midline tracks the inferred median of $N_e\tau$.

DOI: https://doi.org/10.7554/eLife.31257.030

The following source data and figure supplement are available for figure 5:

**Source data 1.** XML to run skygrid analysis on camel-like sequence data and output files.

DOI: https://doi.org/10.7554/eLife.31257.032

**Figure supplement 1.** Skygrid comparison between whole and fragmented genomes.

DOI: https://doi.org/10.7554/eLife.31257.031

currently existing models of population structure (*Vaughan et al., 2014*) can identify distinct demographic modes in MERS-CoV genomic data, where viruses continuously circulating in camels repeatedly jump into humans and cause small outbreaks doomed to extinction (*Figure 1—figure supplement 1*). This inference succeeds under different choices of priors for unknown demographic parameters (*Figure 1—figure supplement 2*) and in the presence of strong biases in sequence sampling schemes (*Figure 3*). When rapid coalescence in the human deme is not allowed (*Figure 1—figure supplement 4*) structured coalescent inference loses power and ancestral state reconstruction is nearly identical to that of discrete trait analysis (*Figure 1—figure supplement 3*). When allowed different deme-specific population sizes, the structured coalescent model succeeds because a large proportion of human sequences fall into tightly connected clusters, which informs a low estimate for the population size of the human deme. This in turn informs the inferred state of long ancestral branches in the phylogeny, that is, because these long branches are not immediately coalescing, they are most likely in camels.

From sequence data, we identify at least 50 zoonotic introductions of MERS-CoV into humans from the reservoir (*Figure 1*), from which we extrapolate that hundreds more such introductions must have taken place (*Figure 3*). Although we recover migration rates from our model (*Figure 1—figure supplement 2*), these only pertain to sequences and in no way reflect the epidemiologically relevant *per capita* rates of zoonotic spillover events. We also looked at potential seasonality in MERS-CoV spillover into humans. Our analyses indicated a period of three months where the odds of a sequenced spillover event are increased, with timing consistent with an enzootic amongst camel calves (*Figure 2*). As a result of our identification of large and asymmetric flow of viral lineages into humans we also find that the basic reproduction number for MERS-CoV in humans is well below the epidemic threshold (*Figure 3*). Having said that, there are highly customisable coalescent methods available that extend the methods used here to accommodate time varying migration rates and population sizes, integrate alternative sources of information and fit to stochastic nonlinear models (*Rasmussen et al., 2014*), which would be more appropriate for MERS-CoV. Some distinct aspects of MERS-CoV epidemiology could not be captured in our methodology, such as hospital outbreaks where $R_0$ is expected to be consistently closer to 1.0 compared to community transmission of MERS-CoV. Outside of coalescent-based models, there are population structure models that explicitly relate epidemiological parameters to the branching process observed in sequence data (*Kühnert et al., 2016*), but often rely on specifying numerous informative priors and can suffer from MCMC convergence issues.

Strong population structure in viruses often arises through limited gene flow, either due to geography (*Dudas et al., 2017*), ecology (*Smith et al., 2009*) or evolutionary forces (*Turner et al., 2005*; *Dudas et al., 2015*). On a smaller scale, population structure can unveil important details about transmission patterns, such as identifying reservoirs and understanding spillover trends and risk, much as we have done here. When properly understood naturally arising barriers to gene flow can be exploited for more efficient disease control and prevention, as well as risk management.

## Transmissibility differences between zoonoses and pandemics

Severe acute respiratory syndrome (SARS) coronavirus, a Betacoronavirus like MERS-CoV, caused a serious epidemic in humans in 2003, with over 8000 cases and nearly 800 deaths. Since MERS-CoV was also able to cause significant pathogenicity in the human host it was inevitable that parallels would be drawn between MERS-CoV and SARS-CoV at the time of MERS discovery in 2012. Although we describe the epidemiology of MERS-CoV from sequence data, indications that MERS-CoV has poor capacity to spread human-to-human existed prior to any sequence data. SARS-CoV swept through the world in a short period of time, but MERS cases trickled slowly and were restricted to the Arabian Peninsula or resulted in self-limiting outbreaks outside of the region, a pattern strongly indicative of repeat zoonotic spillover. Infectious disease surveillance and control measures remain limited, so much like the SARS epidemic in 2003 or the H1N1 pandemic in 2009, zoonotic pathogens with $R_0>1.0$ are probably going to be discovered after spreading beyond the original location of spillover. Even though our results show that MERS-CoV does not appear to present an imminent global threat, we would like to highlight that its epidemiology is nonetheless concerning.

Pathogens *Bacillus anthracis*, Andes hantavirus (*Martinez et al., 2005*), monkeypox (*Reed et al., 2004*) and influenza A are able to jump species barriers but only influenza A viruses have historically resulted in pandemics (*Lipsitch et al., 2016*). MERS-CoV may join the list of pathogens able to jump species barriers but not spread efficiently in the new host. Since its emergence in 2012, MERS-CoV has caused self-limiting outbreaks in humans with no evidence of worsening outbreaks over time. However, sustained evolution of the virus in the reservoir and continued flow of viral lineages into humans provides the substrate for a more transmissible variant of MERS-CoV to possibly emerge. Previous modelling studies (*Antia et al., 2003*; *Park et al., 2013*) suggest a positive relationship between initial $R_0$ in the human host and probability of evolutionary emergence of a novel strain which passes the supercritical threshold of $R_0>1.0$. This leaves MERS-CoV in a precarious position; on one hand its current $R_0$ of $\sim0.7$ is certainly less than 1, but its proximity to the supercritical threshold raises alarm. With very little known about the fitness landscape or adaptive potential of MERS-CoV, it is incredibly challenging to predict the likelihood of the emergence more transmissible variants. In light of these difficulties, we encourage continued genomic surveillance of MERS-CoV in

the camel reservoir and from sporadic human cases to rapidly identify a supercritical variant, if one does emerge.

## Materials and methods

### Sequence data

All MERS-CoV sequences were downloaded from GenBank and accession numbers are given in *Supplementary file 1* (*Assiri et al., 2016a*, *2016b*; *Azhar et al., 2014*; *van Boheemen et al., 2012*; *Briese et al., 2014*; *Chu et al., 2014*; *Cotten et al., 2013*, *2014*; *Drosten et al., 2013*, *2015*; *Fagbo et al., 2015*; *Haagmans et al., 2014*; *Hemida et al., 2014*; *Hunter et al., 2016*; *Kandeil et al., 2016*; *Kapoor et al., 2014*; *Kim et al., 2015*, *2016*; *Lamers et al., 2016*; *Lau et al., 2016*; *Lu et al., 2017*; *Park et al., 2016a*, *2016b*; *Plipat et al., 2017*; *Raj et al., 2014*; *Sabir et al., 2016*; *Seong et al., 2016*; *Xie et al., 2015*). Sequences without accessions were kindly shared by Ali M. Somily, Mazin Barry, Sarah S. Al Subaie, Abdulaziz A. BinSaeed, Fahad A. Alzamil, Waleed Zaher, Theeb Al Qahtani, Khaldoon Al Jerian, Scott J.N. McNabb, Imad A. Al-Jahdali, Ahmed M. Alotaibi, Nahid A. Batarfi, Matthew Cotten, Simon J. Watson, Spela Binter, and Paul Kellam prior to publication. Sequences available on GenBank but not associated with publications were shared by Meriadeg Ar Gouilh, Emad M. Elassal, Monica Galiano, Krista Queen and Ying Tao. Fragments of some strains submitted to GenBank as separate accessions were assembled into a single sequence. Only sequences covering at least 50% of MERS-CoV genome were kept, to facilitate later analyses where the alignment is split into two parts in order to account for effects of recombination (*Dudas and Rambaut, 2016*). Sequences were annotated with available collection dates and hosts, designated as camel or human, aligned with MAFFT (*Katoh and Standley, 2013*), and edited manually. Protein coding sequences were extracted and concatenated, reducing alignment length from 30,130 down to 29,364, which allowed for codon-partitioned substitution models to be used. The final dataset consisted of 174 genomes from human infections and 100 genomes from camel infections (*Supplementary file 1*).

### Phylogenetic analyses

#### Primary analysis, structured coalescent

For our primary analysis, the MultiTypeTree module (*Vaughan et al., 2014*) of BEAST v2.4.3 (*Bouckaert et al., 2014*) was used to specify a structured coalescent model with two demes – humans and camels. At time of writing structured population models are available in BEAST v2 (*Bouckaert et al., 2014*) but not in BEAST v1 (*Drummond et al., 2012*). We use the more computationally intensive MultiTypeTree module (*Vaughan et al., 2014*) over approximate methods also available in BEAST v2, such as BASTA (*De Maio et al., 2015*), MASCOT (*Mueller et al., 2017*), and PhyDyn (*Volz, 2012*). Structured coalescent model implemented in the MultiTypeTree module (*Vaughan et al., 2014*) estimates four parameters: rate of coalescence in human viruses, rate of coalescence in camel viruses, rate of migration from the human deme to the camel deme and rate of migration from the camel deme to the human deme. Analyses were run on codon position partitioned data with two separate HKY+$\Gamma_4$(*Hasegawa et al., 1985*; *Yang, 1994*) nucleotide substitution models specified for codon positions 1 + 2 and 3. A relaxed molecular clock with branch rates drawn from a lognormal distribution (*Drummond et al., 2006*) was used to infer the evolutionary rate from date calibrated tips. Default priors were used for all parameters except for migration rates between demes for which an exponential prior with mean 1.0 was used. All analyses were run for 200 million steps across ten independent Markov chains (MCMC runs) and states were sampled every 20,000 steps. Due to the complexity of multitype tree parameter space 50% of states from every analysis were discarded as burn-in, convergence assessed in Tracer v1.6 and states combined using LogCombiner distributed with BEAST v2.4.3 (*Bouckaert et al., 2014*). Three chains out of ten did not converge and were discarded altogether. This left 70,000 states on which to base posterior inference. Posterior sets of typed (where migrating branches from structured coalescent are collapsed, and migration information is left as a switch in state between parent and descendant nodes) trees were summarised using TreeAnnotator v2.4.3 with the common ancestor heights option (*Heled and Bouckaert, 2013*). A maximum likelihood phylogeny showing just the genetic relationships between MERS-CoV genomes from camels and humans was recovered using PhyML (*Guindon et al., 2003*)

under a HKY+$\Gamma_4$ (*Hasegawa et al., 1985*; *Yang, 1994*) nucleotide substitution model and is shown in *Figure 1—figure supplement 5*.

## Control, structured coalescent with different prior

As a secondary analysis to test robustness to choice of prior, we set up an analysis where we increased the mean of the exponential distribution prior for migration rate to 10.0. All other parameters were identical to the primary analysis and as before 10 independent MCMC chains were run. In this case, two chains did not converge. This left 80,000 states on which to base posterior inference. Posterior sets of typed trees were summarised using TreeAnnotator v2.4.3 with the common ancestor heights option (*Heled and Bouckaert, 2013*).

## Control, structured coalescent with equal deme sizes

To better understand where statistical power of the structured coalescent model lies we set up a tertiary analysis where a model was set up identically to the first structured coalescent analysis, but deme population sizes were enforced to have the same size. This analysis allowed us to differentiate whether statistical power in our analysis is coming from effective population size contrasts between demes or the backwards-in-time migration rate estimation. Five replicate chains were set up, two of which failed to converge after 200 million states. Combining the three converging runs left us with 15,000 trees sampled from the posterior distribution, which were summarised in TreeAnnotator v2.4.3 with the common ancestor heights option (*Heled and Bouckaert, 2013*).

## Control, structured coalescent with more than one tree per genome

Due to concerns that recombination might affect our conclusions (*Dudas and Rambaut, 2016*), as an additional secondary analysis, we also considered a scenario where alignments were split into two fragments (fragment 1 comprised of positions 1–21000, fragment 2 of positions 21000–29364), with independent clocks, trees and migration rates, but shared substitution models and deme population sizes. Fragment positions were chosen based on consistent identification of the region around nucleotide 21000 as a probable breakpoint by GARD (*Kosakovsky Pond et al., 2006*) by previous studies into SARS and MERS coronaviruses (*Hon et al., 2008*; *Dudas and Rambaut, 2016*). All analyses were set to run for 200 million states, subsampling every 20,000 states. Chains not converging after 200 million states were discarded. 20% of the states were discarded as burn-in, convergence assessed with Tracer 1.6 and combined with LogCombiner. Three chains out of ten did not converge. This left 70,000 states on which to base posterior inference. Posterior sets of typed trees were summarised using TreeAnnotator v2.4.3 with the common ancestor heights option (*Heled and Bouckaert, 2013*).

## Control, discrete trait analysis

A currently widely used approach to infer ancestral states in phylogenies relies on treating traits of interest (such as geography, host, *etc.*) as features evolving along a phylogeny as continuous time Markov chains with an arbitrary number of states (*Lemey et al., 2009*). Unlike structured coalescent methods, such discrete trait approaches are independent from the tree (i.e. demographic) prior and thus unable to influence coalescence rates under different trait states. Such models have been used in the past to infer the number of MERS-CoV host jumps (*Zhang et al., 2016*) with results contradicting other sources of information. In order to test how a discrete trait approach compares to the structured coalescent we used our 274 MERS-CoV genome data set in BEAST v2.4.3 (*Bouckaert et al., 2014*) and specified identical codon-partitioned nucleotide substitution and molecular clock models to our structured coalescent analysis. To give the most comparable results, we used a constant population size coalescent model, as this is the demographic function for each deme in the structured coalescent model. Five replicate MCMC analyses were run for 200 million states, three of which converged onto the same posterior distribution. The converging chains were combined after removing 20 million states as burn-in to give a total of 27,000 trees drawn from the posterior distribution. These trees were then summarised using TreeAnnotator v2.4.5 with the common ancestor heights option (*Heled and Bouckaert, 2013*).

## Introduction seasonality

We extracted the times of camel-to-human introductions from the posterior distribution of multitype trees. This distribution of introduction times was then discretised as follows: for sample $k = 1, 2, \ldots, L$ from the posterior, $Z_{ijk}$ was 1 if there as an introduction in month $i$ and year $j$ and 0 otherwise. We model the sum of introductions at month $i$ and year $j$ across the posterior sample $Y_{ij} = \sum_{k=1}^{L} Z_{ijk}$ with the hierarchical model:

$$Y_{ij} \sim \text{Binomial}(L, \theta_{ij})$$
$$\theta_{ij} = \text{inverse logit}(\alpha_j + \beta_i)$$
$$\alpha_j \sim \text{Normal}(\mu_y, \sigma_y)$$
$$\mu_y \sim \text{Normal}(0, 1)$$
$$\sigma_y \sim \text{Cauchy}(0, 2.5)$$
$$\beta_i \sim \text{Normal}(0, \sigma_m)$$
$$\sigma_m \sim \text{Cauchy}(0, 2.5),$$

where $\alpha_j$ represents the contribution of year to expected introduction count and $\beta_i$ represents the contribution of month to expected introduction count. Here, $\text{inverse logit}(\alpha_j + \beta_i) = \frac{\exp(\alpha_j + \beta_i)}{\exp(\alpha_j + \beta_i) + 1}$. We sampled posterior values from this model via the Markov chain Monte Carlo methods implemented in Stan (*Carpenter et al., 2016*). Odds ratios of introductions were computed for each month $i$ as $\text{OR}_i = \exp(\beta_i)$.

## Epidemiological analyses

Here, we employ a Monte Carlo simulation approach to identify parameters consistent with observed patterns of sequence clustering (*Figure 3—figure supplement 2*). Our structured coalescent analyses indicate that every MERS outbreak is a contained cross-species spillover of the virus from camels into humans. The distribution of the number of these cross-species transmissions and their sizes thus contain information about the underlying transmission process. At heart, we expect fewer larger clusters if fundamental reproductive number $R_0$ is large and more smaller clusters if $R_0$ is small. A similar approach was used in *Grubaugh et al. (2017)* to estimate $R_0$ for Zika introductions into Florida.

Branching process theory provides an analytical distribution for the number of eventual cases $j$ in a transmission chain resulting from a single introduction event with $R_0$ and dispersion parameter $\omega$ (*Blumberg and Lloyd-Smith, 2013*). This distribution follows

$$\Pr(j|R_0, \omega) = \frac{\Gamma(\omega j + j - 1)}{\Gamma(\omega j)\,\Gamma(j + 1)} \frac{(\frac{R_0}{\omega})^{j-1}}{(1 + \frac{R_0}{\omega})^{\omega j + j - 1}}. \tag{1}$$

Here, $R_0$ represents the expected number of secondary cases following a single infection and $\omega$ represents the dispersion parameter assuming secondary cases follow a negative binomial distribution (*Lloyd-Smith et al., 2005*), so that smaller values represent larger degrees of heterogeneity in the transmission process.

As of 10 May 2017, the World Health Organization has been notified of 1952 cases of MERS-CoV (*World Health Organization, 2017*). We thus simulated final transmission chain sizes using *Equation 1* until we reached an epidemic comprised of $N = 2000$ cases. 10,000 simulations were run for 121 uniformly spaced values of $R_0$ across the range [0.5–1.1] with dispersion parameter $\omega$ fixed to 0.1 following expectations from previous studies of coronavirus behavior (*Lloyd-Smith et al., 2005*). Each simulation results in a vector of outbreak sizes $\mathbf{c}$, where $c_i$ is the size of the $i$th transmission cluster and $\sum_{i=1}^{K} c_i = 2000$ and $K$ is the number of clusters generated.

Following the underlying transmission process generating case clusters $\mathbf{c}$, we simulate a secondary process of sampling some fraction of cases and sequencing them to generate data analogous to what we empirically observe. We sample from the case cluster size vector $\mathbf{c}$ without replacement according to a multivariate hypergeometric distribution (see Algorithm 1: Multivariate hypergeometric sampling scheme). The resulting sequence cluster size vector $\mathbf{s}$ contains $K$ entries, some of which are zero (i.e. case clusters not sequenced), but $\sum_{i=1}^{K} s_i = 174$ which reflects the number of human

MERS-CoV sequences used in this study. Note that this 'sequencing capacity' parameter does not vary over time, even though MERS-CoV sequencing efforts have varied in intensity, starting out slow due to lack of awareness, methods and materials and increasing in response to hospital outbreaks later. As the default sampling scheme operates under equiprobable sequencing, we also simulated biased sequencing by using concentrated hypergeometric distributions where the probability mass function is squared (bias = 2) or cubed (bias = 3) and then normalized. Here, bias enriches the hypergeometric distribution so that sequences are sampled with weights proportional to $(c_1^{\text{bias}}, c_2^{\text{bias}}, \ldots, c_k^{\text{bias}})$. Thus, bias makes larger clusters more likely to be 'sequenced'.

After simulations were completed, we identified simulations in which the recovered distribution of sequence cluster sizes $\mathbf{s}$ fell within the 95% highest posterior density intervals for four summary statistics of empirical MERS-CoV sequence cluster sizes recovered via structured coalescent analysis: number of sequence clusters, mean, standard deviation and skewness (third central moment). These values were 48–61 for number of sequence clusters, 2.87–3.65 for mean sequence cluster size, 4.84–6.02 for standard deviation of sequence cluster sizes, and 415.40–621.06 for skewness of sequence cluster sizes.

We performed a smaller set of simulations with 2500 replicates and twice the number of cases, that is, $\sum_{i=1}^{K} C_i = 4000$, to explore a dramatically underreported epidemic. Additionally, we performed additional smaller set of simulations on a rougher grid of $R_0$ values (23 values, 0.50–1.05), with 5 values of dispersion parameter $\omega$ (0.002, 0.04, 0.1, 0.5, 1.0) and 3 levels of bias (1,2,3) to justify our choice of dispersion parameter $\omega$ that was fixed to 0.1 in the main analyses (*Figure 3—figure supplement 6*).

## Algorithm 1: Multivariate hypergeometric sampling scheme

Pseudocode describes the multivariate hypergeometric sampling scheme that simulates sequencing. Probability of sequencing a given number of cases from a case cluster depends on cluster size and sequences left (i.e. 'sequencing capacity'). The bias parameter determines how probability mass function of the hypergeometric distribution is concentrated.

**Data:** Array of case cluster sizes in outbreak $\mathbf{c} = (c_1, c_2, \ldots, c_K)$, sequences available $M$, total outbreak size $N$, where $N = \sum_{i=1}^{K} c_i$.

**Result:** Array of sequence cluster sizes sampled: $\mathbf{s} = (s_1, s_2, \ldots, s_K)$.

Draw $s_i$ from a hypergeometric distribution with $c_i$ successes, $N - c_i$ failures after $M$ trials;

**while** $i < K$ **do**

$i = i + 1$;

$M = M - s_{i-1}$;

Compute the probability mass function (pmf) for all possible values of $s_i$,

$\mathbf{p} = (p(0)^{\text{bias}}, p(1)^{\text{bias}}, \ldots, p(c_i)^{\text{bias}}) \times (\sum_i p_i^{\text{bias}})^{-1}$, where $p(\cdot)$ is the pmf for a hypergeometric distribution with $c_i$ successes, $N - c_i$ failures after $M$ trials;

Draw a sequence cluster size $s_i$ from array of potential sequence cluster sizes $(0, 1, \ldots, c_i)$ according to $\boldsymbol{p}$;

**end**

Add remaining sequences to last sequence cluster $c_K = M - s_{K-1}$;

## Demographic inference of MERS-CoV in the reservoir

In order to infer the demographic history of MERS-CoV in camels we used the results of structured coalescent analyses to identify introductions of the virus into humans. The oldest sequence from each cluster introduced into humans was kept for further analysis. This procedure removes lineages coalescing rapidly in humans, which would otherwise introduce a strong signal of low effective population size. These subsampled MERS-CoV sequences from humans were combined with existing sequence data from camels to give us a dataset with minimal demographic signal coming from epidemiological processes in humans. Sequences belonging to the outgroup clade where most of MERS-CoV sequences from Egypt fall were removed out of concern that MERS epidemics in Saudi Arabia and Egypt are distinct epidemics with relatively poor sampling in the latter. Were more sequences of MERS-CoV available from other parts of Africa we speculate they would fall outside of the diversity that has been sampled in Saudi Arabia and cluster with early MERS-CoV sequences from Jordan and sequences from Egyptian camels. However, currently there are no indications of

what MERS-CoV diversity looks like in camels east of Saudi Arabia. A flexible skygrid tree prior (*Gill et al., 2013*) was used to recover estimates of scaled effective population size ($N_e\tau$) at 50 evenly spaced grid points across six years, ending at the most recent tip in the tree (2015 August) in BEAST v1.8.4 (*Drummond et al., 2012*), under a relaxed molecular clock with rates drawn from a lognormal distribution (*Drummond et al., 2006*) and codon position partitioned (positions $1+2$ and $3$) HKY $+\Gamma_4$ (*Hasegawa et al., 1985*; *Yang, 1994*) nucleotide substitution models. At time of writing advanced flexible coalescent tree priors from the skyline family, such as skygrid (*Gill et al., 2013*) are available in BEAST v1 (*Drummond et al., 2012*) but not in BEAST v2 (*Bouckaert et al., 2014*). We set up five independent MCMC chains to run for 500 million states, sampling every $50\,000$ states. This analysis suffered from poor convergence, where two chains converged onto one stationary distribution, two to another and the last chain onto a third stationary distribution, with high effective sample sizes. Demographic trajectories recovered by the two main stationary distributions are very similar and differences between the two appear to be caused by convergence onto subtly different tree topologies. This non-convergence effect may have been masked previously by the use of all available MERS-CoV sequences from humans which may have lead MCMC towards one of the multiple stationary distributions.

To ensure that recombination was not interfering with the skygrid reconstruction, we also split our MERS-CoV alignment into ten parts 2937 nucleotides long. These were then used as separate partitions with independent trees and clock rates in BEAST v1.8.4 (*Drummond et al., 2012*). Nucleotide substitution and relaxed clock models were set up identically to the whole genome skygrid analysis described above (*Drummond et al., 2006*; *Hasegawa et al., 1985*; *Yang, 1994*). Skygrid coalescent tree prior (*Gill et al., 2013*) was used jointly across all ten partitions for demographic inference. Five MCMC chains were set up, each running for 200 million states, sampling every 20,000 states.

### Data availability

Sequence data and all analytical code is publicly available at https://github.com/blab/mers-structure (*Dudas, 2017*). A copy is archived at https://github.com/elifesciences-publications/mers-structure.

# Acknowledgements

We would like to thank Allison Black for useful discussion and advice. GD is supported by the Mahan postdoctoral fellowship from the Fred Hutchinson Cancer Research Center. TB is a Pew Biomedical Scholar and is supported by NIH R35 GM119774-01. AR was supported in part by the European Union Seventh Framework Programme for research, technological development and demonstration under Grant Agreement no. 278433-PREDEMICS and no. 725422-RESERVOIRDOCS, and the Wellcome Trust through project 206298/Z/17/Z.

We gratefully acknowledge the contribution of the following scientists for sharing MERS-CoV sequence data before publication:

Ali M. Somily[1], Mazin Barry[1], Sarah S. Al Subaie[1], Abdulaziz A. BinSaeed[1], Fahad A. Alzamil[1], Waleed Zaher[1], Theeb Al Qahtani[1], Khaldoon Al Jerian[1], Scott J.N. McNabb[2], Imad A. Al-Jahdali[3], Ahmed M. Alotaibi[4], Nahid A. Batarfi[5], Matthew Cotten[6], Simon J. Watson[6], Spela Binter[6], Paul Kellam[6].

[1]College of Medicine, King Saud University, Riyadh, Kingdom of Saudi Arabia [2]Rollins School of Public Health, Emory University, Atlanta, GA, USA [3]Deputy Minister. Ex. General Director King Fahad General Hospital, Jeddah and Occupational and environmental medicine, Um AlQura University, Kingdom of Saudi Arabia [4]Department of Intensive Care, Prince Mohammed bin Abdulaziz Hospital, Riyadh, Kingdom of Saudi Arabia [5]Epidemiology section, Command and Control Center (CCC) Ministry of Health, Jeddah [6]Wellcome Trust Sanger Institute, Hinxton, United Kingdom

# Additional information

### Funding

| Funder | Grant reference number | Author |
| --- | --- | --- |
| National Institutes of Health | R35 GM119774-01 | Trevor Bedford |
| Pew Charitable Trusts | Pew Biomedical Scholar | Trevor Bedford |

| European Commission | 278433-PREDEMICS | Andrew Rambaut |
| Wellcome | 206298/Z/17/Z | Andrew Rambaut |
| Fred Hutchinson Cancer Research Center | Mahan Postdoctoral Fellowship | Gytis Dudas |
| European Commission | 725422-RESERVOIRDOCS | Andrew Rambaut |

The funders had no role in study design, data collection and interpretation, or the decision to submit the work for publication.

### Author contributions

Gytis Dudas, Conceptualization, Data curation, Software, Formal analysis, Funding acquisition, Validation, Investigation, Visualization, Methodology, Writing—original draft, Writing—review and editing; Luiz Max Carvalho, Software, Formal analysis, Investigation, Methodology, Writing—review and editing; Andrew Rambaut, Conceptualization, Data curation, Supervision, Funding acquisition, Methodology, Writing—review and editing; Trevor Bedford, Conceptualization, Resources, Supervision, Funding acquisition, Methodology, Project administration, Writing—review and editing

### Author ORCIDs

Gytis Dudas http://orcid.org/0000-0002-0227-4158
Trevor Bedford https://orcid.org/0000-0002-4039-5794

### Decision letter and Author response

Decision letter https://doi.org/10.7554/eLife.31257.037
Author response https://doi.org/10.7554/eLife.31257.038

## Additional files

### Supplementary files

• Source data 1. MERS-CoV sequences used in the study.
DOI: https://doi.org/10.7554/eLife.31257.033

• Supplementary file 1. Strain names, accessions (where available), identified host and reported collection dates for MERS-CoV genomes used in this study.
DOI: https://doi.org/10.7554/eLife.31257.034

• Transparent reporting form
DOI: https://doi.org/10.7554/eLife.31257.035

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
