## [Decision Letter]

Thank you for submitting your article "MERS-CoV spillover at the camel-human interface" for consideration by *eLife*. Your article has been reviewed by three peer reviewers, and the evaluation has been overseen by a Reviewing Editor and Prabhat Jha as the Senior Editor. The following individuals involved in review of your submission have agreed to reveal their identity: Erik Volz (Reviewer #1); Christophe Fraser (Reviewer #3).

The reviewers have discussed the reviews with one another and the Reviewing Editor has drafted this decision to help you prepare a revised submission.

Summary:

In this paper, Dudas et al. perform a coalescent analysis of 274 MERS-Cov viruses. They conclude that 1) MERS is sustained in camels (R0>1) and not in humans (R0<1), and that most if not all cross species transmissions have been camel to human. 2) Cross species events are seasonal, but R0 in humans isn't. 3) The relatively low levels of genetic diversity in camel viruses can be explained by camel demography.

Essential revisions:

1) The population genetic model (the particular form of structured coalescent) is highly idealised and this may influence the quantitative conclusions, although we suspect the conclusions are quite robust qualitatively. This model specifically estimates the rate of a lineage moving between demes going backwards in time; the numbers cited for the camel->human rate is really the rate that a lineage in humans goes to a camel going down the tree. The relationship between these migration rates and the epidemiologically meaningful transmission rate is complex and depends among other things on the ratio of population sizes in both demes. Per-capita transmission rates could be estimated using an epidemiologically structured coalescent model (see e.g. papers by Volz and Rasmussen), which would ideally be stochastic due to bursty dynamics in humans. But this would be a large undertaking and so we suggest that for now the distinction is clarified. Overall, a little more discussion of the complexity and pitfalls when relating idealised population genetic models (like the island model used here) to a noisy nonlinear epidemic like this one might be merited.

2) 'Our analyses recover these results despite sequence data heavily skewed towards non-uniformly sampled human cases and are robust to choice of prior.' This is a quite nice result and raises the question if skewed sampling would bias estimates if using a substitution model approach ('discrete trait analysis', DTA). It would strengthen the paper to include a comparison of the structured coalescent estimates to another method for ancestral states; the most popular approach in beast has been substitution models (DTA). These may give divergent results because of skewed sampling. It would be rather easy for the authors to run a DTA and if biased, this would serve as a good cautionary example when sampling is highly skewed towards one deme.

3) A comparison to ML tree reconstruction could potentially be illuminating. We think you could be clearer about what drives the results in your paper. It is unusual for a phylogenetic ancestral reconstruction, that the results seem to be determined as much by the coalescent assumptions as by the tree topology. The two-patch model had a much higher coalescent rate in the human deme than in the camel deme – so long branches are only really possible in the camel deme. This may be why for example, staring at the top clade of Figure 1, one can see camel ancestry to a whole bunch of human sequences that are not topologically separated by camel sequences. If this is correct, these results may not necessarily be wrong, but it made us slightly uncomfortable that the results are driven by the coalescent model, not the tree topology. Please elaborate, either correcting us, or explaining better. A simple test of this hypothesis would be that an ML ancestral reconstruction on the ML tree would not give the same clusters. I don't think that would make the ML result correct, but it might be an enlightening comparison. Or you may prefer another way to address this.

4) Easily addressed, but important. The paper already sounds a strong voice of concern in the final paragraph, but we think this could be even stronger. Antia et al. Nature 2003 first showed, using a simple branching process, that for most genetic landscapes, the probability of a pathogen evolving to state with R0>1 increases dramatically as a function of the wild-type R0. So R0~0.8 is much worse than R0~0.3. More sophisticated models have been done since, especially by Llyod-smith's group, but the basic result is sound. In the light of this theoretical work, your findings are not at all reassuring.

5) More generally, the model choices need better explaining. Why delve into a structured coalescent in BEAST2 for the ancestral reconstruction, but go back to the Skygrid in BEAST1 for computations of Ne? We assume this is a pragmatic choice, and for the latter you carefully reduced the human clusters to reduce bias, but we think the rationale for your choices need laying out more clearly. Even if pragmatic rather than principled, (e.g. there are no structure coalescent options in BEAST1), we think it still needs to be stated why you made the choices you did. Especially since there are other recently-developed BEAST2 packages that could be used to fit the same structured coalescent model: BASTA and MASCOT, as well as the very flexible PhyDyn package (which might offer improvements in computation time).

---

## [Author Response]

Essential revisions:1) The population genetic model (the particular form of structured coalescent) is highly idealised and this may influence the quantitative conclusions, although we suspect the conclusions are quite robust qualitatively. This model specifically estimates the rate of a lineage moving between demes going backwards in time; the numbers cited for the camel->human rate is really the rate that a lineage in humans goes to a camel going down the tree. The relationship between these migration rates and the epidemiologically meaningful transmission rate is complex and depends among other things on the ratio of population sizes in both demes. Per-capita transmission rates could be estimated using an epidemiologically structured coalescent model (see e.g. papers by Volz and Rasmussen), which would ideally be stochastic due to bursty dynamics in humans. But this would be a large undertaking and so we suggest that for now the distinction is clarified. Overall, a little more discussion of the complexity and pitfalls when relating idealised population genetic models (like the island model used here) to a noisy nonlinear epidemic like this one might be merited.

Yes, we agree that the structured coalescent approach is idealised and does not reflect a meaningful rate of zoonotic transfer of lineages, which is the reason we restricted any mention of rates to supplementary figures and do not attach numbers whenever rates are mentioned, but still report the number of introductions observed in the sequence data. We have altered Figure 1—figure supplement 2 in the reviewed manuscript to reflect that the rates shown are backwards in time. We have added the following sentences to the Discussion to highlight the fact that the coalescent model employed is not ideal:

“Although we recover migration rates from our model (Figure 1—figure supplement 2), these only pertain to sequences and in no way reflect the epidemiologically relevant per capita rates of zoonotic spillover events. […] Outside of coalescent-based models there are population structure models that explicitly relate epidemiological parameters to the branching process observed in sequence data (Kuhnert et al., 2016), but often rely on specifying numerous informative priors and can suffer from MCMC convergence issues.”

2) 'Our analyses recover these results despite sequence data heavily skewed towards non-uniformly sampled human cases and are robust to choice of prior.' This is a quite nice result and raises the question if skewed sampling would bias estimates if using a substitution model approach ('discrete trait analysis', DTA). It would strengthen the paper to include a comparison of the structured coalescent estimates to another method for ancestral states; the most popular approach in beast has been substitution models (DTA). These may give divergent results because of skewed sampling. It would be rather easy for the authors to run a DTA and if biased, this would serve as a good cautionary example when sampling is highly skewed towards one deme.

An excellent suggestion, thank you. We have run this analysis and include the results as a new figure supplement (Figure 1—figure supplement 3). As expected, the skewed sampling results in a reconstruction of ancestral states that puts humans as the source of most MERS-CoV lineages in camels. We have added the appropriate description of methods as well as the following paragraph in Results:

“Our findings suggest that instances of human infection with MERS-CoV are more common than currently thought, with exceedingly short transmission chains mostly limited to primary cases that might be mild and ultimately not detected by surveillance or sequencing. […] We suspect that this particular discrete trait analysis reconstruction is false due to biased data, i.e. having nearly twice as many MERS-CoV sequences from humans (n = 174) than from camels (n = 100) and the inability of the model to account for and quantify vastly different rates of coalescence in the phylogenetic vicinity of both types of sequences.”

3) A comparison to ML tree reconstruction could potentially be illuminating. We think you could be clearer about what drives the results in your paper. It is unusual for a phylogenetic ancestral reconstruction, that the results seem to be determined as much by the coalescent assumptions as by the tree topology. The two-patch model had a much higher coalescent rate in the human deme than in the camel deme – so long branches are only really possible in the camel deme. This may be why for example, staring at the top clade of Figure 1, one can see camel ancestry to a whole bunch of human sequences that are not topologically separated by camel sequences. If this is correct, these results may not necessarily be wrong, but it made us slightly uncomfortable that the results are driven by the coalescent model, not the tree topology. Please elaborate, either correcting us, or explaining better. A simple test of this hypothesis would be that an ML ancestral reconstruction on the ML tree would not give the same clusters. I don't think that would make the ML result correct, but it might be an enlightening comparison. Or you may prefer another way to address this.

A very good point. We share the suspicion that the results are largely driven by contrasts in effective population sizes between demes. In addition to the requested maximum likelihood phylogeny (now Figure 1—figure supplement 5) we also ran a structured coalescent analysis where deme sizes are enforced to be the same (now Figure 1—figure supplement 4). This model fails in a similar way to a DTA reconstruction shown in the Figure 1—figure supplement 3. We now explain how the structured coalescent arrives at the tree shown in Figure 1 in the Discussion:

“When allowed different deme-specific population sizes, the structured coalescent model succeeds because a large proportion of human sequences fall into tightly connected clusters, which informs a low estimate for the population size of the human deme. This in turn informs the inferred state of long ancestral branches in the phylogeny, i.e. because these long branches are not immediately coalescing, they are most likely in camels.”

4) Easily addressed, but important. The paper already sounds a strong voice of concern in the final paragraph, but we think this could be even stronger. Antia et al. Nature 2003 first showed, using a simple branching process, that for most genetic landscapes, the probability of a pathogen evolving to state with R0>1 increases dramatically as a function of the wild-type R0. So R0~0.8 is much worse than R0~0.3. More sophisticated models have been done since, especially by Llyod-smith's group, but the basic result is sound. In the light of this theoretical work, your findings are not at all reassuring.

We agree that this is an important point, but also feel that it is difficult to formulate warnings about pandemic potential without overstating the case. We also believe that adaptive landscapes play a considerable role in emerging pandemics. We added an additional sentence to the Discussion and refer to the Antia et al. study:

“Previous modeling studies (Antia et al., 2003; Park et al., 2013) suggest a positive relationship between initial R0 in the human host and probability of evolutionary emergence of a novel strain which passes the supercritical threshold of R0 > 1.0. […] In light of these difficulties, we encourage continued genomic surveillance of MERS-CoV in the camel reservoir and from sporadic human cases to rapidly identify a supercritical variant, if one does emerge.

5) More generally, the model choices need better explaining. Why delve into a structured coalescent in BEAST2 for the ancestral reconstruction, but go back to the Skygrid in BEAST1 for computations of Ne? We assume this is a pragmatic choice, and for the latter you carefully reduced the human clusters to reduce bias, but we think the rationale for your choices need laying out more clearly. Even if pragmatic rather than principled, (e.g. there are no structure coalescent options in BEAST1), we think it still needs to be stated why you made the choices you did. Especially since there are other recently-developed BEAST2 packages that could be used to fit the same structured coalescent model: BASTA and MASCOT, as well as the very flexible PhyDyn package (which might offer improvements in computation time).

In hindsight we see how our choices may have seemed arbitrary. Indeed all cases of using BEAST 1 vs. BEAST 2 came down to what models were implemented in which package. We have clarified our choices throughout the manuscript.